# The Putative Virulence Plasmid pYR4 of the Fish Pathogen *Yersinia ruckeri* Is Conjugative and Stabilized by a HigBA Toxin–Antitoxin System

**DOI:** 10.3390/biology13090652

**Published:** 2024-08-23

**Authors:** Fisentzos Floras, Chantell Mawere, Manvir Singh, Victoria Wootton, Luke Hamstead, Gareth McVicker, Jack C. Leo

**Affiliations:** Antimicrobial Resistance, Omics and Microbiota Group, Department of Biosciences, Nottingham Trent University, Nottingham NG1 4FQ, UK

**Keywords:** conjugation, toxin–antitoxin system, virulence plasmid, *Yersinia ruckeri*

## Abstract

**Simple Summary:**

The bacterium *Yersinia ruckeri* causes enteric redmouth disease, which affects salmonid fish, including farmed fish such as Atlantic salmon and rainbow trout, leading to significant commercial losses. In this study, we investigated a plasmid (an independently replicating DNA molecule) called pYR4 that is suspected of aiding *Y. ruckeri* in causing disease by producing adhesive structures known as type 4 pili. We identified a toxin–antitoxin locus on the plasmid, *higBA*, which we hypothesized might prevent loss of the plasmid from the bacterial cell through “addiction” to the antitoxin HigA. HigA is quickly degraded if the plasmid is lost, whereas the toxin HigB is stable and leads to cell death in the absence of HigA. We were able to demonstrate that HigBA is a functional toxin–antitoxin system and that stabilizes pYR4. We further showed that pYR4 can transfer between bacteria by a process known as conjugation. However, loss of the pYR4 plasmid did not have an effect in a simple insect larva infection model. Further work will have to be undertaken to evaluate the role of pYR4 in enteric redmouth disease. These results will help in understanding and ultimately managing a serious disease of commercially important fish.

**Abstract:**

The bacterium *Yersinia ruckeri* causes enteric redmouth disease in salmonids and hence has substantial economic implications for the farmed fish industry. The Norwegian *Y. ruckeri* outbreak isolate NVH_3758 carries a relatively uncharacterized plasmid, pYR4, which encodes both type 4 pili and a type 4 secretion system. In this study, we demonstrate that pYR4 does not impose a growth burden on the *Y. ruckeri* host bacterium, nor does the plasmid contribute to twitching motility (an indicator of type 4 pilus function) or virulence in a *Galleria mellonella* larval model of infection. However, we show that pYR4 is conjugative. We also reveal, through mutagenesis, that pYR4 encodes a functional post-segregational killing system, HigBA, that is responsible for plasmid maintenance within *Y. ruckeri*. This is the first toxin–antitoxin system to be characterized for this organism. Whilst further work is needed to elucidate the virulence role of pYR4 and whether it contributes to bacterial disease under non-laboratory conditions, our results suggest that the plasmid possesses substantial stability and transfer mechanisms that imply importance within the organism. These results add to our understanding of the mobile genetic elements and evolutionary trajectory of *Y. ruckeri* as an important commercial pathogen, with consequences for human food production.

## 1. Introduction

Farming of aquatic organisms such as fish and shellfish, or aquaculture, is an important and growing industry to meet the world’s nutritional needs. In 2021, the amount of fish produced through aquaculture equaled the amount of fish caught from wild fisheries [1], and the proportion of farmed fish is now exceeding that of wild-captured fish [2]. However, the crowded and stressful conditions in fish farms predispose the livestock to infections by bacterial, viral, and parasitic pathogens [3]. Infections in aquaculture have been estimated to cost the global economy over USD 6 billion per year [4].

One such fish pathogen is the bacterium *Yersinia ruckeri*, the causative agent of enteric redmouth disease (ERM). ERM affects mainly salmonid fish, including commercially important species such as Atlantic salmon and rainbow trout [5]. Though ERM has been controlled to some extent through vaccination, outbreaks caused by bio-serotypes not covered or poorly controlled by existing vaccines have occurred in recent years [6,7,8]. In addition, the pathogenesis of ERM is not well understood. Although several virulence factors of *Y. ruckeri* have been identified [9], their role in the disease has not been fully characterized. Therefore, investigation into the disease mechanisms of *Y. ruckeri* is important for understanding its pathogenesis and developing novel control measures.

The human pathogenic *Yersinia* species rely on plasmid-mediated virulence to cause disease [10]. Similarly, most *Y. ruckeri* isolates contain plasmids [11,12]. Recently, the plasmid pYR4 was discovered in the Norwegian outbreak strain NVH_3758, encoding a type 4 pilus (TFP) cluster and a type 4 secretion system (T4SS) [13]. Type 4 pili are contractile fimbrial adhesins that mediate twitching motility but are also often involved in biofilm formation, pathogenesis, or DNA uptake [14]. T4SSs (distinct structures from TFP) include conjugation apparatus [15], which is why pYR4 is presumed conjugative [13]. The *tra* locus of *Y. ruckeri* 150R, encoding a T4SS, has been implicated in virulence [16]. The region encoding this T4SS locus as well as a *pil* locus encoding TFP is highly similar to regions on the plasmids pYR3 and pYR4 and so is also likely plasmid-borne [13]. Due to the presence of the TFP and T4SS loci on pYR4, it has been suspected to be a virulence plasmid. Efforts to cure pYR4 were unsuccessful [13], suggesting it is stably maintained in the population.

Bioinformatic examination of pYR4 revealed a toxin–antitoxin system locus, *higBA*. Toxin–antitoxin (TA) systems, also known as addiction factors, encode a stable toxin and an antitoxin that is relatively unstable. If the antitoxin gene is present, the antitoxin pool can be regenerated, and the toxin is not able to exert its toxicity on the cell. However, if the antitoxin gene is lost (e.g., through curing of the plasmid encoding it), the stable toxin persists while any remaining antitoxin is degraded, leading to toxic effects resulting in cessation of growth or death of the cell. Therefore, the host bacteria become “addicted” to the plasmid because its maintenance is essential for a healthy growing population [17,18]. The HigBA module is part of a wider group of TA systems known as type II TA systems, in which both the toxin and antitoxin are proteins, and the antitoxin binds directly to and sequesters the toxin [19]. The *higBA* operon is unusual in that the toxin gene is transcribed upstream of the antitoxin, contrary to most type II TA operons [20]. The HigB protein itself is a ribosome-dependent mRNA endoribonuclease and is a member of the RelE family of toxins [21].

We hypothesized that the HigBA TA system of pYR4 contributes to its stability. In this study, we demonstrate that the *higBA* locus is a functional TA system in *Y. ruckeri* and that its deletion results in a reduction in the stability of pYR4. This is, to our knowledge, the first characterization of a TA system in *Y. ruckeri*. We further provide evidence to show that pYR4 is a conjugative plasmid. However, using a simple infection model, larvae of the greater wax moth (*Galleria mellonella*), we did not observe any difference in virulence between the wild-type and the plasmid-cured strains. The status of pYR4 as a virulence plasmid thus remains unresolved.

## 2. Materials and Methods

### 2.1. Bioinformatics

Based on the previous annotation of pYR4 [13], genes encoding putative TA or partitioning systems were manually verified by additional BLASTP [22] analysis of the translated protein products. Promoter sequences were predicted using the Softberry-BPROM programme v. 1.0 [23]. Putative conjugation systems were predicted using MOB-Typer v. 3.1.9 [24].

### 2.2. Bacterial Strains and Growth Conditions

The bacterial strains used in this study were *Escherichia coli* DH5α and *Y. ruckeri* NVH_3758 [25,26]. Bacteria were grown in lysogeny broth (LB) [27] and supplemented with antibiotics where necessary; ampicillin at 100 µg/mL, chloramphenicol at 25 µg/mL, and neomycin and streptomycin at 50 µg/mL. 1.5% (*w*/*v*) agar was added to prepare solid media. For experiments in minimal medium, M9 was used [28], either with glycerol at 0.2% rather than glucose, or supplemented with iron(II) sulfate (10 µM), thiamine (5 µg/mL), BME vitamins (Merck KGaA, Darmstadt, Germany), and A5 trace metals (Merck). *E. coli* was propagated at 37 °C and *Y. ruckeri* at 30 °C unless otherwise indicated.

### 2.3. Cloning

For toxicity assays, the *higB* gene was amplified using *Y. ruckeri* NVH_3758 genomic DNA as the template using Q5 polymerase (New England Biolabs, Ipswich, MA, USA). Whole genome DNA was extracted from *Y. ruckeri* using the Wizard^®^ Genomic DNA Purification Kit (Promega, Madison, WI, USA). Primers (Table A1 in Appendix A) were designed to include overhangs for Gibson assembly [29] of the insert into the arabinose-inducible vector pBAD33 [30]. To produce the antitoxin construct, the entire operon (*higBA*, including upstream, and downstream regions) was amplified as above with overhangs complementary to the pGM101 vector [31]. Alternatively, variants of the *higBA* operon were cloned using mutagenic primers to produce pGM101::Δ*higB* (lacking the *higB* gene entirely) or pGM101::*higB*_mut_ (containing a *higB* start codon ATG → CTG mutation). Vectors were linearized by PCR and digested with DpnI to remove the circular template. The constructs were then assembled using the NEBuilder HiFi DNA Assembly kit (New England Biolabs), and the reaction mixtures were transformed into *E. coli* DH5α. Transformants were selected for using chloramphenicol (pBAD33) or ampicillin (pGM101) and insert-positive clones were screened for using colony PCR. All constructs were verified by Sanger sequencing.

### 2.4. Creating a higB Knockout and Introducing a Selection Marker into pYR4

To knock out the *higB* gene in *Y. ruckeri*, the upstream (US) and downstream (DS) regions (600 bp in length) of the *higBA* operon were amplified by PCR. In addition, the *higBA* promoter-*higA* sequence was amplified from pGM101::Δ*higB* as well as a chloramphenicol resistance cassette (*cat* gene plus its promoter and flanking flippase recognition target sites) from pKD3 [32]. Primers (Table A1 in Appendix A) were designed such that these PCR products could be annealed through Gibson assembly to yield US-*higA*-*cat*-DS as a single DNA fragment lacking the *higB* gene (Figure A1a in Appendix A). This fragment was produced using the NEBuilder HiFi assembly kit, and the assembled reaction was used as a template for amplifying the US-*higA*-*cat*-DS fragment. To generate the knockout, the λ Red helper plasmid pMJH65 [33] was transformed into NVH_3758 by electroporation, and transformants were selected for on LB + ampicillin. A transformant colony was grown in LB + ampicillin at 30 °C and the λ Red recombination proteins were induced by the addition of 0.2% arabinose for an hour before harvesting the cells for electroporation. Then, 1 µg of the US-*higA*-*cat*-DS fragment was used for electroporation; the recovery medium in which the bacteria were incubated for 2 h at 30 °C was supplemented with magnesium and arabinose (LB + 10 mM MgCl_2_ + 0.2% arabinose). The bacteria were then plated on LB + chloramphenicol. Transformants were screened by PCR to identify clones with either insertion of the *cat* gene after the intact *higBA* operon (pYR4::*cat higB*^+^) or deletion of *higB* and insertion of the *cat* gene after *higA* (pYR4::*cat* Δ*higB*) (Figure A1b in Appendix A). pMJH65 was removed by growing the bacteria overnight at 37 °C without selection and plating for single colonies, followed by screening for ampicillin-sensitive clones. The correctness of the insertions (either *cat* or Δ*higB cat*) was verified by Sanger sequencing of the amplified *hig* locus. pMJH65 was a gift from Mark Liles (Addgene plasmid #67273; http://n2t.net/addgene:67273, accessed on 13 June 2024; RRID:Addgene_67273).

### 2.5. Plasmid Curing and Plasmid Loss Experiments

To cure pYR4, we initially grew bacteria at an elevated temperature (37 °C) for extended periods and tested for the presence of pYR4 by PCR. However, this did not result in any observable plasmid loss, and neither did transient repeated heat shocks at higher temperatures (up to 47 °C). Plasmid curing only occurred after the introduction of the pGM101::Δ*higB* into NVH_3758 by electroporation and subsequent culture of a confirmed transformant at 37 °C in LB + ampicillin, followed by testing for the presence of the plasmid by PCR. Once a pYR4-negative clone had been verified, the pGM101::Δ*higB* plasmid was cured by culturing in the absence of ampicillin overnight, diluting the culture 1:10,000 the following day in fresh medium, and reculturing, over the course of three days. Then the culture was plated for single colonies and ampicillin-sensitive clones were screened for by streaking individual colonies first on LB + ampicillin and then LB; clones that grew on LB but not LB + ampicillin were chosen. That these were *Y. ruckeri* was confirmed by 16S rDNA PCR.

To determine the effect of the *higB* deletion, the NVH_3758 strains with pYR4::*cat* derivatives (either *higB*^+^ or Δ*higB*) were inoculated in biological quadruplicate in LB and grown overnight at 37 °C to stress the cells and expedite plasmid loss. The following day, the cultures were diluted 1:10,000 in fresh medium and regrown at 37 °C. This was continued for a total of 10 days. To quantify plasmid loss, dilution series of the cultures were plated onto LB with no selection and LB + chloramphenicol; only bacteria still retaining the plasmid would grow on the selection plate. We then calculated the percentage of resistant colonies in the overall viable count of the cultures.

### 2.6. Conjugation

To prepare recipient cells for conjugation experiments, spontaneous streptomycin-resistant mutants were isolated by plating cultures of *Y. ruckeri* NVH_3758 pYR4^−^ onto LB medium with 50 µg/mL of streptomycin. After two days of culture at 30 °C, colonies were restreaked onto streptomycin medium to verify the resistance. Alternatively, we introduced a neomycin resistance marker-containing plasmid (pGM101_neo_) by electroporation and selection on neomycin (50 µg/mL). This plasmid was determined to be stably kept in the population during overnight culture even in the absence of selection. For conjugation, the streptomycin or neomycin-resistant pYR4^-^ strains were used as the recipient, mixed with the pYR4::*cat higB*^+^ donor. The strains were grown in liquid medium separately overnight with selection, and then 1 mL of the cultures were pelleted at 3000× *g* for 5 min, washed, and then resuspended in 500 µL PBS. The donor and recipient strains were then mixed at different ratios (1:1, 2:1, 1:2) in a total of 100 µL of PBS, and this was spotted onto an LB plate without selection and grown overnight at 30 °C. The bacteria were then scraped off the plate, diluted in PBS, and plated for single colonies on counterselection plates containing chloramphenicol to select for the pYR4::*cat* plasmid and neomycin/streptomycin to select for the recipient, then grown overnight at 30 °C. Putative transconjugants from the chloramphenicol-neomycin plates were screened by PCR for both the pYR4 plasmid and the pGM101_neo_ plasmid.

### 2.7. Toxicity Assays

Bacterial viability in response to toxin–antitoxin production was assayed according to [31]. Briefly, cells containing pBAD33::*higB* together with either pGM101, pGM101::*higBA*, pGM101::Δ*higB*, or pGM101::*higB*_mut_ were grown overnight in LB + ampicillin + chloramphenicol + 0.2% glucose to repress toxin expression. Cultures were then subcultured and grown to an optical density at 600 nm (OD_600nm_) = 0.1 in fresh media at 37 °C, before pelleting and resuspending the bacteria in prewarmed LB + ampicillin + chloramphenicol + 1% arabinose to induce toxin gene expression. At 0, 15, 30, 60, and 180 min post-induction, samples were serially diluted in PBS before plating onto LB agar + ampicillin + chloramphenicol + 0.2% glucose for quantification of the number of viable colony-forming units (CFU) at each timepoint.

### 2.8. Twitching Motility Assays

Twitching motility was assayed using a macroscopic twitching assay, essentially as described in [34]. Briefly, bacteria were stabbed into the center of a 1% (*w*/*v*) LB agar plate down to the agar-plastic interface. The plates were then incubated overnight at 25 °C, 30 °C, and 37 °C, with three biological triplicates for all temperatures, after which the interstitial halo was measured (no developer was added). As a positive control, we used *Pseudomonas aeruginosa* PA14.

### 2.9. Galleria Infection Assays

For in vivo infection assays, we used *Galleria mellonella* larvae as a model (purchased from LiveFoods Direct, Sheffield, UK). Upon arrival, the larvae were checked and any blackening or pupating larvae were discarded. The larvae were then weighed and ones between 200 and 300 mg were used in the experiments. For infections, *Y. ruckeri* NVH_3758 pYR4^+^ and pYR4^−^ strains were cultured overnight at 30 °C; the following day, subcultures were prepared and grown till an OD_600nm_ of 0.4–0.5. These cultures were then diluted to specific CFU/mL in PBS, based on a standard curve relating CFU/mL to OD_600nm_. In addition to live bacteria, we also included heat-killed NVH_3758 pYR4^+^ (20 min at 80 °C) as a control to test whether bacterial cell components posed any toxicity to the larvae. Then, 10 *Galleria* were injected with each sample using 27 G needles and 1 mL syringes with a mechanical syringe pump (Cole-Parmer, St Neots, UK) at a rate of 13.21 mL/h and a volume of 10 µL per injection. Injections were made on the bottom left proleg into the hemocoel of the larvae. Inoculum doses were checked by diluting and plating from 10 µL drops from the syringe. Sterile PBS-injected and uninjected larvae were used as negative controls. Finally, 5 larvae from each group were incubated in Petri dishes at either 30 °C or 25 °C for 96 h. Scoring of the larvae’s health was performed according to the scale of Serrano et al. [35], with higher scores indicating healthy larvae.

### 2.10. Statistical Analysis

Statistical analyses were carried out using GraphPad Prism v9. Two-way ANOVA followed by Tukey’s, Dunnett’s, or Sidak’s multiple comparisons tests were performed to analyze data as appropriate. Specific tests are noted in the figure legends. Statistical significance was assumed if *p* < 0.05.

## 3. Results

### 3.1. pYR4 Carries Several Putative TA-Related Genes

The 80.8 kb putative virulence plasmid pYR4 from *Y. ruckeri* strain NVH_3758 (Figure 1a) contains genes potentially encoding a T4SS and TFP [13]. In addition, a manual analysis of pYR4 identified several sequences of interest for plasmid stabilization, such as genes potentially encoding two partitioning systems (StbAB and ParAG) and TA systems. TA systems and TA-related/orphaned proteins encoded within the plasmid’s so-called “stability cluster” include a putative type IV system toxin (CbtA) and its associated XRE-family antidote, lone antitoxins from the Phd, MqsA, and ParD families and an Abi-family protein. The plasmid also encodes an intact HigBA type II TA system; pYR4 HigB is 39.25% identical (50.46% similar) to the HigB-2 toxin of the *Vibrio cholerae* N16961 superintegron, whereas pYR4 HigA is 23.46% identical (37.75% similar) to the *Vibrio* HigA-2 antitoxin [21]. Interestingly, the *cbtA* gene is truncated by an IS6-family transposase in the pYR4-like plasmid pYR3 found in *Y. ruckeri* strain CSF007-82 (Figure 1b), removing the first 29 amino acids at the protein’s N-terminus and presumably the gene’s promoter, and creating a region of sequence diversity between the two plasmids upstream of the *cbtA* pseudogene. We therefore focused our investigation on the conserved *higBA* locus, which is shared between pYR3 and pYR4.

### 3.2. pYR4 HigBA Is a Functional Post-Segregational Killing System

In order to ascertain whether pYR4 *higBA* encodes a bona fide toxin–antitoxin system, both toxin and antitoxin activity were assessed. The toxin gene *higB* and its native ribosome binding site were cloned into the arabinose-inducible promoter on pBAD33, creating pHigB. Due to the inverted nature of the operon, three antitoxin-expressing constructs (Figure 2a) were cloned separately into compatible vector pGM101 to test their ability to abrogate toxicity: one containing the entire wild-type operon (pGM101::*higBA*), one containing the operon with a deletion of the *higB* gene (pGM101::Δ*higB*) and one containing the entire operon with an A → C mutation in the start codon of *higB* (pGM101::*higB*_mut_), preventing its translation. Each antitoxin construct included the operon’s native promoter in order to take advantage of conditional cooperativity [36,37].

Expression of the toxin gene from pBAD33 in *E. coli* DH5α in the absence of antitoxin resulted in approximately 100-fold loss of cell viability at 180 min post-induction relative to strains carrying any of the three plasmid variants encoding the antitoxin (Figure 2b; *p* = 0.0003). Curing pYR4 from NVH_3758 (see below) allowed us to also test the effect of the TA system in its native host. Here, the effect of the toxin was far more dramatic (10,000-fold reduction in viability) but was still abolished by any of the three antitoxin constructs (Figure 2c; *p* < 0.0001). Hence, HigBA encoded on pYR4 is a functional and cognate toxin–antitoxin pair.

In order to test pYR4 stability under laboratory conditions, we grew *Y. ruckeri* NVH_3758 for several weeks with daily subculturing into fresh media. At various timepoints during the experiment, cultures were serially diluted and plated to obtain single colonies, then those colonies were tested for the presence of pYR4 using a multiplex PCR targeting key plasmid genes *repA*, *pilN*, and *higB* (Figure A2 in Appendix A). No evidence of plasmid loss was observed. The same was true when cultures were briefly heat-shocked to induce stress-based plasmid loss at temperatures up to 47 °C, the maximum temperature for which we obtained viable bacteria after recovery and plating. pYR4 seems, therefore, to be highly stable, in agreement with previous observations [13].

Given that we had previously shown that HigA is able to prevent HigB toxicity and hence interfere with TA system function (Figure 2), we transformed NVH_3758 with pGM101::*higB*_mut_ and tested plasmid loss in the resulting transformant cells using the multiplex PCR as before. Results showed approximately 30% of colonies lacked *repA* and *pilN* (though retained the *higB* band, as this was present on the antitoxin vector; Figure A3 in Appendix A) within 10 days of subculture, sharply contrasting our attempts at plasmid curing without the use of HigA. Subsequent growth in non-selective medium resulted in spontaneous curing of the pGM101::*higB*_mut_ vector (assessed by PCR and ampicillin sensitivity), resulting in NVH_3758_pYR4^−^.

Lastly, as direct proof of TA-mediated plasmid addiction, we used λ Red recombineering to construct a Δ*higB* deletion at the native pYR4 locus tagged with a chloramphenicol resistance marker, and the corresponding chloramphenicol-resistant *higB*^+^ control strain. After 10 daily subcultures, pYR4Δ*higB* retention (measured by chloramphenicol resistance) was only 41% of that of the *higB*^+^ plasmid (Figure 3; *p* < 0.0001). These results, alongside the relative ease of pYR4 curing via the pGM101::*higB*_mut_ construct, are consistent with HigBA being a functional post-segregational killing system that enhances the stability of pYR4.

### 3.3. Qualitative Evidence of pYR4 Conjugation

Since pYR4 putatively encodes a T4SS, we surmised it may be conjugative. Analysis via MOB-Typer [24] failed to identify any conjugation machinery but suggested that the plasmid may be mobilizable; however, since MOB-Typer is by the authors’ own admission focussed on Enterobacteriaceae plasmids, its lack of recognition of *Yersinia* mobilization/conjugation elements is not surprising.

In order to experimentally test the conjugative capacity of pYR4, we employed the *higB*^+^ chloramphenicol-resistant variant produced previously in this work and tested its ability to transfer into either a recipient strain harboring pGM101_neo_, conferring neomycin resistance, or a recipient strain harboring a spontaneous streptomycin resistance mutation generated through growth on streptomycin in our laboratory. In both cases, results were impossible to quantify due to an extremely high level of background growth post-conjugation. However, larger colonies were identified as transconjugants by PCR, containing both pYR4 and pGM101_neo_ in the relevant experiment, therefore giving qualitative evidence of successful conjugation (Figure 4). Notably, such colonies continued to grow healthily and provide positive PCR results for both recipient and donor genes when they were restreaked onto fresh medium and re-analyzed, whereas colonies taken from background growth did not grow well after restreaking.

### 3.4. Virulence-Related Phenotypes Encoded by pYR4

We assessed the burden of carrying pYR4 in various laboratory media but found no difference between NVH_3758 and NVH_3758_pYR4^−^ in either growth rate or yield, regardless of media (Figure A4 in Appendix A).

Due to the presence of the putative TFP encoded by pYR4, we assessed the twitching motility of *Y. ruckeri* NVH_3758 and NVH_3758_pYR4^−^ on solid media at a range of temperatures. Results (Figure 5) showed poor motility of *Y. ruckeri* at all temperatures under these experimental conditions regardless of the presence or absence of pYR4 (*p* > 0.95), whilst a *Pseudomonas aeruginosa* PA14 positive control was highly motile (*p* < 0.0001 compared with both *Y. ruckeri* strains).

Lastly, we assessed the contribution of pYR4 to *Y. ruckeri* virulence by comparing NVH_3758 and NVH_3758_pYR4^−^ in a *Galleria mellonella* infection assay at a range of temperatures and infectious doses. Results (Figure 6) showed that *G. mellonella* larvae succumbed rapidly to *Y. ruckeri* infection at all temperatures and doses tested, regardless of the presence or absence of pYR4 (*p* ≥ 0.8234 between the two score curves in Figure 6; survival curves shown in Figure A5 in Appendix A). On the contrary, larvae left uninjected or those injected with PBS only or heat-killed *Y. ruckeri* remained healthy (vs. pYR4^+^: *p* ≤ 0.001 under all conditions).

## 4. Discussion

In this study, we investigated the role of the *higBA* locus in maintaining the plasmid pYR4 in the *Y. ruckeri* strain NVH_3758. We demonstrated that HigBA is a functional TA system that contributes directly to the stability of pYR4, as deletion of this system led to significant loss of the plasmid in a relatively short time (6 days, representing approximately 60 generations). Furthermore, providing the *higA* antitoxin gene in trans in pYR4^+^ cells allowed efficient curing of the plasmid; something that had failed during several previous efforts, both in this study and a previous one [13]. However, neutralizing the HigB toxin only led to ~50% loss of the plasmid even after a 10-day incubation, suggesting other components also contribute to the stability of pYR4. One of these could be the intact CbtA TA system also identified on pYR4; future work will determine its role in stabilizing pYR4 and any compound effects this may have with the HigBA system.

HigBA was originally characterized as a plasmid maintenance element in *Proteus* spp. [38] but has since been found in a wide range of pathogenic bacterial species including *V. cholerae* [21] and both enterohaemorrhagic and uropathogenic *E. coli* [39]. HigB is part of the RelE superfamily of endoribonuclease toxins and is encoded upstream of its cognate antitoxin, which is unusual for type II TA systems [20]. This gene organization is conserved on pYR4 and the related *Y. ruckeri* plasmid, pYR3. Type II TA systems, including HigBA, autoregulate operon expression via binding of the TA complex to their own promoters, in a process known as conditional cooperativity [36,37]. We cloned *higA* under the control of its own promoter and showed that this construct was able to abrogate the toxicity of HigB produced from an induced pBAD33 promoter, suggesting that this regulation mechanism remains intact for the pYR4 HigBA module.

The *higBA* locus is part of a ‘stability cluster’ found in pYR4, and our re-analysis of this region shows that some of this cluster, including *higBA*, is shared with pYR3. Therefore, it is reasonable to assume that also pYR3 is stabilized by *higBA*. The *cbtA* type IV TA locus we identified is also shared between these two plasmids, but it is intact only in pYR4, which bolsters the argument for *higBA* being the main stabilizing factor of both plasmids. The stability of other *Y. ruckeri* plasmids has not been investigated; it is therefore not known whether the high stability of pYR4 (and presumably pYR3) mediated by *higBA* is unusual among *Y. ruckeri* plasmids, or if the others are similarly stable through some other mechanisms. Of note, pYR3 and pYR4 also encode one and two putative partitioning systems, respectively, within their stability clusters. These may contribute to plasmid maintenance, though the systems differ between the two plasmids.

Based on the presence of a T4SS, pYR4 was previously presumed to be conjugative [13]. However, the related *tra* operon was also implicated in virulence in a different strain of *Y. ruckeri*, and deletion of *traI* led to a reduction in virulence in a fish model [16]. The pYR4 T4SS has similarities to the Icm/Dot T4SS of *Legionella pneumophila*, which translocates effector proteins into eukaryotic host cells [40]. The pYR4 *tra* thus encodes an ‘expanded’ T4SS, which is generally virulence-related but can also be conjugation systems [15]. We provide evidence here that pYR4 is a conjugative plasmid. We were able to observe the transfer of antibiotic resistance gene-tagged pYR4 into a pYR4-negative recipient. Though we could not quantify the rate of transfer due to background growth, regardless of multiple attempts with different recipient strain markers, we could identify genuine transconjugants by PCR. These transconjugants were able to grow well when restreaked on counterselection medium, in contrast to PCR-negative colonies on the original counterselection plates. This demonstrates the transfer of the plasmid and strongly suggests that the T4SS mediates conjugation rather than virulence-related effector delivery, being the only potential conjugation apparatus encoded on pYR4. pYR4 does not encode any obvious effector proteins, further strengthening the argument that the T4SS is purely conjugative [13]. However, we cannot currently conclusively rule out whether the T4SS translocates effectors encoded on the chromosome.

If the T4SS is not virulence-related, this suggests that the TFP encoded by an adjacent locus might be responsible for the reduction in virulence observed in a previous study [16]. In our study, we did not observe twitching motility, but this is not the only function attributed to TFP. The pYR4-encoded *pil* locus could be involved in biofilm formation, adhesion to particular surfaces, or possibly aid T4SS-mediated conjugation by bringing bacterial cells into close proximity. TFP has been demonstrated to promote conjugation for some plasmids and other mobile genetic elements [41,42,43]. It is also possible that we did not test for twitching motility under the right conditions. We tested three temperatures, two of which are close to the optimum growth temperature of 28 °C but higher than the temperatures in which *Y. ruckeri* generally causes disease in fish, in waters below 20 °C [44]. Previously, the T4SS of *Y. ruckeri* 150R was found to be more highly expressed at 18 °C compared with 28 °C [16], and more recently higher expression was observed for many other genes at 18 °C [45]. Therefore, it is possible that the TFP may be active in generating twitching motility at lower temperatures than those we tested.

To test the hypothesis of pYR4 being a virulence plasmid, we performed infection experiments using the *Galleria mellonella* larvae model. This model has been used before for *Y. ruckeri* [46]. Previous experiments were performed at the mammalian body temperature of 37 °C, higher than the temperature optimum of *Y. ruckeri*. In our experiments, we tested lower temperatures, 30 °C and 25 °C. At these temperatures, ≥80% of larvae succumbed within 48 h of infection, even with a dose (2 × 10^3^ CFU) lower than one that resulted in only ~20% mortality after 96 h at 37 °C [46]. This suggests that *Y. ruckeri* is indeed more virulent at lower temperatures, as proposed before [45]. We did not observe major differences in overall health scores (Figure 6) or mortality rates (Figure A5) between the tested temperatures. Nonetheless, our results suggest that if the *G. mellonella* model is used in the future for assessing *Y. ruckeri* virulence, temperatures below 30 °C should be preferred.

Contrary to our hypothesis, we did not see any effect of curing pYR4 on virulence. Again, this could be because we tested virulence at a suboptimal temperature, and the role of this plasmid would only become evident at even lower temperatures. A more probable reason is that the *G. mellonella* model is a very crude one, with bacteria injected directly into the hemocoel of the larvae. This is a poor mimic for the infection route in the natural fish host, where the bacteria probably enter through the gills [5], and the *G. mellonella* model may bypass early steps in the infection process where pYR4 plays a role. Therefore, to conclusively determine whether pYR4 and, by extension, pYR3 and others are virulence plasmids, experiments should be conducted in salmonid fish.

## 5. Conclusions

We have shown that the *higBA* locus of pYR4 encodes a functional TA system, the first to be characterized for *Y. ruckeri*. This system contributes to the high stability of this plasmid and deleting the toxin component *higB* or adding the *higA* antitoxin in trans allowed curing the plasmid efficiently. We further provided qualitative evidence to show that pYR4 is conjugative, suggesting the T4SS is involved in horizontal gene transfer rather than virulence. Surprisingly, deletion of pYR4 did not show a significant effect in *Y. ruckeri* virulence in the *Galleria mellonella* model. This may be because the model is not a good mimic for the natural fish host, or other conditions such as temperature were not optimal. Further research is needed to identify the role, if any, of pYR4 and other *Y. ruckeri* plasmids in pathogenesis.

## Figures and Tables

**Figure 1 biology-13-00652-f001:**
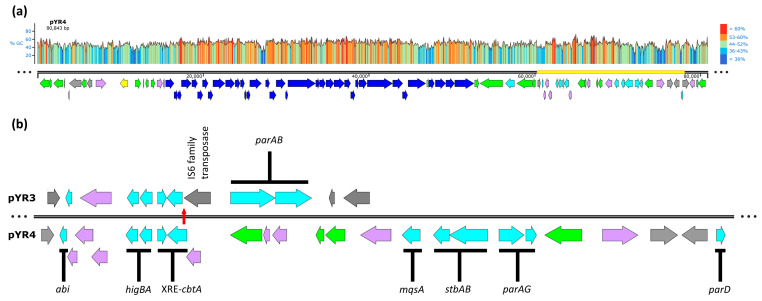
Map of pYR4 and its stability cluster. Predicted gene functions are indicated by colors: dark blue, TFP, and T4SS; gray, mobile genetic elements, and insertion sequences; yellow, plasmid replication; cyan, stability elements; green, genes of other predicted function; lavender, hypothetical ORFs of unknown function. (**a**) Linearized map of pYR4. GC content (%) is shown above by color (right) and graph height (left). Yellow bar shows the region expanded in the lower panel. (**b**) Expanded map of the ~18 kb stability cluster of pYR4 and its comparison to pYR3, indicating individual putative plasmid maintenance elements. Truncation of the *cbtA* gene in pYR3 (top) relative to pYR4 (bottom), and hence the start of sequence divergence of the plasmids, is marked by a vertical red arrow. Alignment anchored on the *higBA* locus. pYR3 ORFs beyond the putative stability cluster are not shown.

**Figure 2 biology-13-00652-f002:**
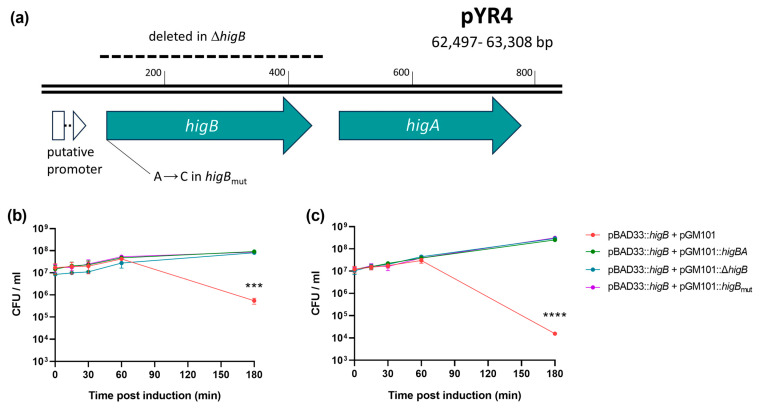
pYR4 encodes a functional toxin–antitoxin, HigBA. (**a**) Map of the 812 bp region included in pGM101::*higBA* and its derivatives, indicating the deletion and mutation in pGM101::Δ*higB* and pGM101::*higB*_mut_, respectively. A putative σ^70^ promoter was predicted by SoftBerry BPROM [23]. (**b**,**c**) Viability of bacteria after induction of *higB* expression from pBAD33 in the presence of antitoxin-encoding vectors or empty vector control, as indicated, in either (**b**) *E. coli* DH5α or (**c**) *Y. ruckeri* NVH_3758_pYR4^−^. *** *p* < 0.001; **** *p* < 0.0001; overall effect of strain calculated by two-way ANOVA (n = 3 biological replicates). Bars show standard error of the mean.

**Figure 3 biology-13-00652-f003:**
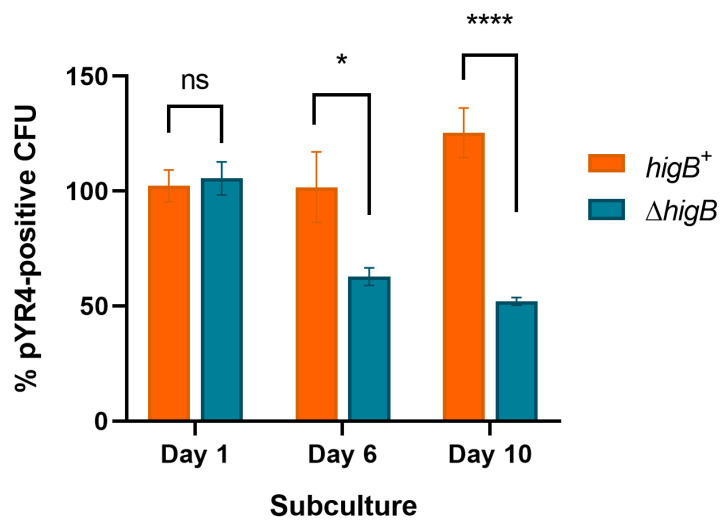
Plasmid loss of pYR4 encoding (*higB*^+^) or lacking (Δ*higB*) a functional HigBA TA system. Per cent plasmid-containing cells after the indicated number of daily subcultures in liquid media, calculated as the proportion of chloramphenicol-resistant versus total CFU grown on solid media. ns, not significant; * *p* < 0.05; **** *p* < 0.0001 by two-way ANOVA with Sidak’s multiple comparisons tests (*n* ≥ 3 biological replicates). Error bars show standard error of the mean.

**Figure 4 biology-13-00652-f004:**
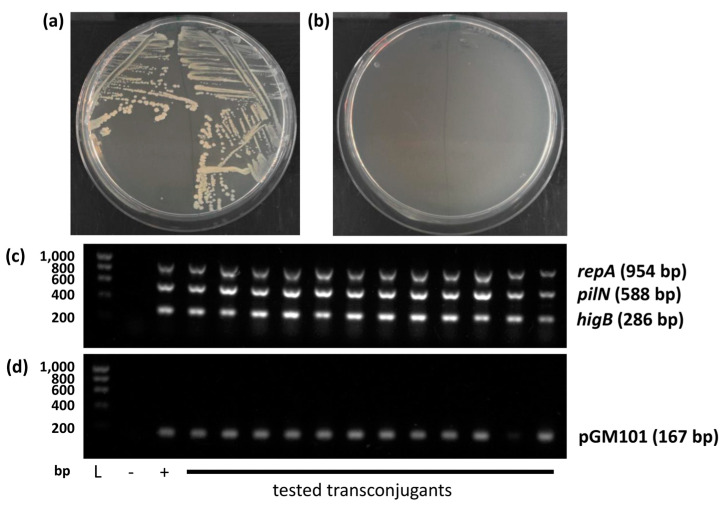
Qualitative evidence of pYR4 conjugation. (**a**) Restreak of two representative transconjugant colonies. (**b**) Restreak of two representative background colonies. (**c**,**d**) PCR to detect (**c**) pYR4 and (**d**) pGM101_neo_. L: Ladder; size (bp) shown left. Amplicon sizes shown right. + and − in (**c**) are pre-conjugation donor and recipient colonies, respectively. + and − in (**d**) are pre-conjugation recipient and donor colonies, respectively.

**Figure 5 biology-13-00652-f005:**
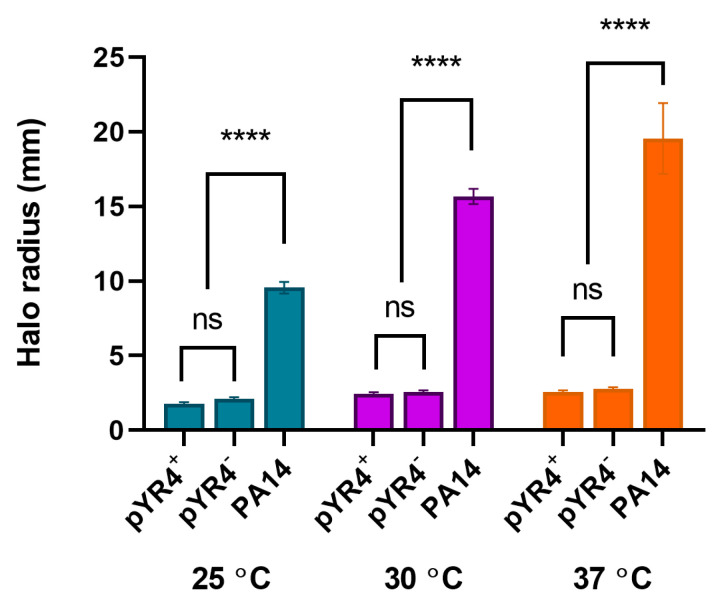
Twitching motility of *Y. ruckeri* carrying (+) or not carrying (−) pYR4**.** Motility “halo” from stab point on solid media measured at the indicated temperatures after growth for 24 h. *P. aeruginosa* PA14 included as a highly motile control. ns, not significant; **** *p* < 0.0001 by two-way ANOVA with Tukey’s multiple comparisons tests (*n* = 3 biological replicates). Error bars show standard error of the mean.

**Figure 6 biology-13-00652-f006:**
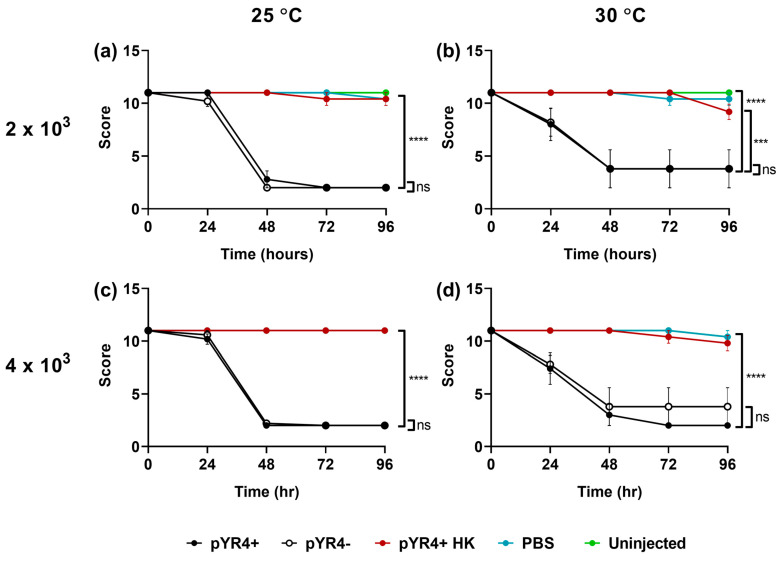
The impact of pYR4 in a *Galleria mellonella* model of infection. Larvae were left uninjected or were injected with 2 × 10^3^ (**a**,**b**) or 4 × 10^3^ (**c**,**d**) CFU *Y. ruckeri* NVH_3758 (“pYR4^+^”) or NVH_3758_pYR4^–^ (“pYR4^−^”), or an equivalent volume of PBS or heat-killed *Y. ruckeri* (“pYR4^+^ HK”), as indicated. Larvae were incubated at either 25 °C (**a**,**c**) or 30 °C (**b**,**d**) for 96 h and their health scored every 24 h according to the Galleria scoring system by Serrano et al. [35]. Bars show standard error of the mean. Statistical analysis by two-way ANOVA with Dunnett’s post test: ns *p* ≥ 0.05; *** *p* < 0.001; **** *p* < 0.0001. Assay conducted on *n* = 5 larvae per experimental condition.

## Data Availability

All data are presented in the paper.

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
