# Peer review of "The Putative Virulence Plasmid pYR4 of the Fish Pathogen Yersinia ruckeri Is Conjugative and Stabilized by a HigBA Toxin–Antitoxin System"

_biology, 2024, doi:10.3390/biology13090652_

Round 1

Reviewer 1 Report

Comments and Suggestions for Authors

1) A separate statistics section should be included in the materials and methods

2) In Figure 6, it is not clear to which points reliable differences refer. Why did the authors not conduct statistical comparisons between the experimental groups?

3) The authors should correct the text layout in the article

Author Response

We thank the Reviewer for their comments to improve the manuscript.

Comment 1: A separate statistics section should be included in the materials and methods

Response 1: We have added a section on statistics to the Methods (lines 292-296).

Comment 2: In Figure 6, it is not clear to which points reliable differences refer. Why did the authors not conduct statistical comparisons between the experimental groups?

Response 2: We did in fact statistically analyse some of the individual experimental groups previously, which is how we were able to report the P value comparing pYR4+ vs pYR4- in the original manuscript. We have now extended this analysis to all groups and have revised both the figure and text to reflect and clarify this.

Comment 3: The authors should correct the text layout in the article

Response 3: It is not clear to us what the reviewer is referring to. We have used the template provided by Biology, and as far as we can tell, the layout follows the template.

Reviewer 2 Report

Comments and Suggestions for Authors

The bacterium Yersinia ruckeri affects salmonid fish, causing enteric redmouth disease and leading to significant commercial losses. In this article, authors investigated the Y. ruckeri pYR4 plasmid, which is suspected of aiding the bacteria to cause disease. The article shows that pYR4 encodes a functional post-segregational killing system, TA system HigBA, responsible for pYR4 plasmid maintenance within Y. ruckeri. In addition, the authors demonstrated that pYR4 is conjugative and it does not contribute to twitching motility or virulence in a Gallleria mellonella animal model. The article is novel as is the first TA system to be characterized for this organism and reveal new aspect of the relatively uncharacterized pYR4 plasmid.

Major comments:

1.        L416-417 and L423: The authors did not provide sufficient evidence to conclude this. They did not have experiments showing that the pYR4 T4SS have a role in conjugation or virulence, instead they show this for the entire plasmid.

2.        Section 3.1. “pYR4 carries several putative TA-related genes”. The authors do not detail the method used to identify TA systems. This information needs to be detailed in Materials and Methods and explained at results section.

Minor comments:

L75: “antitoxin that is degraded quickly” is not completely true. Not all antitoxins are unstable. Need to be checked.

L229: What it means “manual analysis”? Please to clarify.

L269: “three antitoxin variants”, please clarify. They are not antitoxin variants; antitoxin gene remains intact in those constructions.

L331-334: How the authors differentiated between transconjugant colonies and background colonies? Need to be explained.

L449: incorrect figure citation, it should be Figure A5.

Author Response

We thank the Reviewer for their positive evaluation and comments to improve the manuscript.

Comment 1: L416-417 and L423: The authors did not provide sufficient evidence to conclude this. They did not have experiments showing that the pYR4 T4SS have a role in conjugation or virulence, instead they show this for the entire plasmid.

Response 1: While we agree with the Reviewer that we have not directly demonstrated the pYR4 T4SS is conjugative, this secretion system is the only plausible conjugation apparatus encoded on this plasmid. However, we have amended the wording in this section (lines 511, 518-519) to reflect this.

Comment 2: Section 3.1. “pYR4 carries several putative TA-related genes”. The authors do not detail the method used to identify TA systems. This information needs to be detailed in Materials and Methods and explained at results section.

Response: We have added a section (lines 140-145) on bioinformatic analysis in the Methods to describe the information the Reviewer requests.

Minor comments:

1. L75: “antitoxin that is degraded quickly” is not completely true. Not all antitoxins are unstable. Need to be checked.

Response 1: We disagree with the reviewer. In order to function as a post-segregational killing system, the antitoxin must be less stable than the toxin. We have changed the description to 'relatively unstable' on line 76 to reflect this.

2. L229: What it means “manual analysis”? Please to clarify.

Response 2: We have clarified this in the new section on bioinformatics (lines 140-145).

3. L269: “three antitoxin variants”, please clarify. They are not antitoxin variants; antitoxin gene remains intact in those constructions.

Response 3: Thank you for pointing this out. We have rephrased to "three plasmid variants encoding the antitoxin" on line 345.

4. L331-334: How the authors differentiated between transconjugant colonies and background colonies? Need to be explained.

Response 4: The true transconjugant colonies tended to be larger and were able to grow when restreaked on a counterselection plate. Final identification was done by PCR (as shown in Figure 4). We have rephrased lines 412-413, which is now hopefully clearer.

5. L449: incorrect figure citation, it should be Figure A5.

Response 5: Thank you for noticing. In fact, the first figure reference on this line was incorrect as well. The correct references are Fig. 6 and Fig. A5, rather than figures 5 and A4. We have corrected this on line 563.

Reviewer 3 Report

Comments and Suggestions for Authors

There are very minor grammatical, punctuations, etc. There may be need for brief elaborations of some concepts and need to check whether some acronyms are defined or not. The sequence of the putative promoter sequences should have been searched for consensus signature sequences.

Comments on the Quality of English Language

The quality of English language is very good, except for minor grammatical and punctuation concerns.

Author Response

We thank the Reviewer for their positive evaluation and comments to improve the manuscript.

Comment 1: There are very minor grammatical, punctuations, etc. There may be need for brief elaborations of some concepts and need to check whether some acronyms are defined or not. The sequence of the putative promoter sequences should have been searched for consensus signature sequences.

Response 1: We have gone through the manuscript and tried to eliminate any remaining mistakes. We have also checked that all acronyms (except very standard ones like DNA and PCR) have been defined. Regarding the putative promoter, this was searched for using consensus sequences with the BPROM software. This has now been clarified in the legend (line 358-9) and in a new section on bioinformatics (lines 140-145).